# Cattle Horn Shavings: A Possible Nitrogen Source for Apple Trees

**Juozas Lanauskas [1,*], Nobertas Uselis [1], Loreta Buskienė [1], Romas Mažeika [2], Gediminas Staugaitis [2] and Darius Kviklys [1,3]**

[1]  Lithuanian Research Centre for Agriculture and Forestry, Institute of Horticulture, Kaunas st 30, 54333 Babtai, Lithuania; nobertas.uselis@lammc.lt (N.U.); loreta.buskiene@gmail.com (L.B.); darius.kviklys@lammc.lt (D.K.)

[2]  Agrochemical Research Laboratory, Lithuanian Research Centre for Agriculture and Forestry, Savanoriai ave. 287, 50127 Kaunas, Lithuania; romas.mazeika@lammc.lt (R.M.); gediminas.staugaitis@lammc.lt (G.S.)

[3]  Department of Horticulture, Norwegian Institute of Bioeconomy Research—NIBIO Ullensvang, Ulensvangvegen 1005, NO-5781 Lofthus, Norway

*  Correspondence: juozas.lanauskas@lammc.lt

**Abstract:** The circular economy concept promotes the recycling of agricultural waste. This study was aimed at investigating the effects of cattle horn shavings on apple tree nitrogen nutrition. Ligol apple trees on P 60 rootstock were the object of the study. The experiment was conducted in the experimental orchard of the Institute of Horticulture, Lithuanian Research Centre for Agriculture and Forestry, from 2015 to 2018. Two fertiliser rates were tested: 50 and 100 kg/ha N. Horn shavings (14.1% N) were applied at the end of autumn or at the beginning of vegetation in the spring and in one treatment 100 kg/ha N rate was divided into two equal parts and applied both in autumn and spring. The effects of the horn shavings were compared with the effects of ammonium nitrate (34.4% N) and the unfertilised treatment. The lowest mineral nitrogen content was found in the unfertilised orchard soil and the soil fertilised with horn shavings in the spring at 50 kg/ha N equivalent. In all other cases, the fertilisers increased the soil's mineral nitrogen content. The lowest leaf nitrogen content was found in apple trees that grew in the unfertilised orchard soil or soil fertilised in the spring with 50 kg/ha N of horn shavings (1.58–2.13%). In other cases, leaf nitrogen content was higher (1.77–2.17%). The apple trees with the lowest leaf nitrogen content produced the smallest average yield (34.5–36.6 t/ha). The highest yield was recorded from fruit trees fertilised with 50 kg/ha N of ammonium nitrate applied in spring or horn shavings applied in autumn (42.4 and 41.4 t/ha, respectively). The influence of horn shavings on the other studied parameters was similar to that of ammonium nitrate. Horn shavings, like nitrogen fertiliser, could facilitate nitrogen nutrition management in apple trees, especially in organic orchards, where the use of synthetic fertilisers is prohibited.

**Keywords:** leaf nitrogen; mineral nitrogen; soil; yield

## 1. Introduction

Nitrogen plays an important role in apple tree production [1], and apple tree nitrogen nutrition depends on soil and climate conditions. Soil with high organic matter content, favourable weather conditions and proper soil management enhance the likelihood of adequate nitrogen nutrition in fruit trees [2–4]. If fruit trees cannot satisfy their physiological needs with soil nitrogen, fertilisers are used.

Plants can easily utilise mineral forms of nitrogen. Ammonium or nitrate fertilisers dissolve in the soil solution, and plants subsequently absorb them. They are extremely useful for the rapid correction of nitrogen deficiency in the soil. Apple trees require more nitrogen during the first half of vegetation, so spring application of ammonium nitrate is

the most common practice for enhancing the fruit trees' supply of nitrogen in conventional orchards [5]. Precise nitrogen management using organic fertilisers involves an additional challenge, as the organic fertilisers are usually not as well defined and predictable as mineral ones [6]. Various organic materials are used for nitrogen content enhancement in the soil: compost, manure, manure-based fertilisers, green manures with legumes, etc. [7,8]. The nitrogen in most organic sources is in the form of amino acids or proteins. It becomes available to plants only during the process of mineralisation. Biotic and abiotic factors influence this process. In practice, it is difficult to predict the course of mineralisation and control the amount of mineral nitrogen released during the vegetation period [9].

Cattle horn shavings, which contain about 15% of nitrogen (N), constitute a promising source of nitrogen fertiliser [10]. In addition, they are biodegradable waste from the meat industry. The advantage of cattle horn shavings over other organic fertilisers is considerably higher nitrogen content. Protein hydrolysates from bovine horns and hooves also have the potential to be used for foliar fertilisation, whether alone or in a combination with other complementary hydrolysates [11].

According to the requirements of EU regulations [12], animal by-products can be used in the organic farming system. Products of animal origin are used to fertilize plants, including keratin-containing waste: horn chips, shredded feathers, bristles, horn core powder, etc. [13,14]. Keratin contains about 15–18% nitrogen, 1.5–2.0% phosphorus, sulphur and other elements that are in organic form. Keratin-containing wastes decompose more slowly than other organic animal wastes because of cysteine, sulphur-containing amino acids, that form strong intermolecular bonds and give to the protein a crystalline structure and strength. Cattle horns used to be a great raw material for haberdashery, but recently they have been replaced by more colourful and easier to recycle plastics.

More than half of the animal by-products are not suitable for normal consumption, because of their unusual physical and chemical characteristics [15] Jayathilakan et al., 2012. Slaughterhouses generate between 0.5 and 2 kg of horns and other keratin-containing waste from each slaughtered bovine animal. As a result, thousands of tons of waste are accumulated each year that could be used in organic horticulture.

Cattle horn shavings could be beneficial for optimising nitrogen nutrition in fruit trees. However, data on the effects of these fertilisers remain limited.

Therefore, this research is aimed at evaluating the possibility of using cattle horn shavings as nitrogen fertiliser for apple trees.

## 2. Materials and Methods

The trial was conducted in the experimental orchard of the Institute of Horticulture (55°60′ N, 23°48′ E), at the Lithuanian Research Centre for Agriculture and Forestry from 2015 to 2018. The experiment was performed in 2–6-year-old apple tree orchard with 3.5 × 1.25 m planting scheme. Apple trees of the Ligol cultivar on P 60 rootstock were investigated. The experimental plots were fully randomised and had four replications. Each experimental plot contained five fruit trees, and guard trees were planted at the ends. Orchard floor management comprised frequently mown grass in the alleyways with 1.5 m wide herbicide strips along the tree rows. The orchard was not irrigated, and fruitlets were not thinned.

Scheme of the experiment:

1.    No fertilisers
2.    50 kg/ha N equivalent applied in spring as $NH_4NO_3$
3.    100 kg/ha N equivalent applied in spring as $NH_4NO_3$
4.    50 kg/ha N equivalent applied in spring as horn shavings
5.    100 kg/ha N equivalent applied in spring as horn shavings
6.    50 + 50 N equivalent applied at equal parts in autumn and spring as horn shavings
7.    50 kg/ha N equivalent applied in autumn as horn shavings
8.    100 kg/ha N equivalent applied in autumn as horn shavings

In spring fertilizers were applied at 07–09 BBCH, in autumn—at 95 BBCH stage [16]. The first autumn application of horn shavings was performed in 2014. The effects of the horn shavings were compared to the effects of ammonium nitrate (34.4% N) and unfertilised treatment. Both fertilisers were broadcasted throughout the entire area of the relevant experimental plots on the soil surface.

Cattle horn shavings from a haberdashery company were used. The waste generated during production is shredded into horn shavings with a special mill. A 2.5–3.0 mm chip fraction was used for the research, which is similar in size to the mineral fertilizer granules, containing 14.1% N (Table 1).

**Table 1.** Chemical composition of cattle horn shavings.

| Composition | Content, % |
|---|---|
| Dry matter | 91.20 ± 1.60 |
| Organic matter | 98.30 ± 0.90 |
| Organic carbon | 38.65 ± 1.30 |
| Total nitrogen (N): <br> N in organic form, <br> N in inorganic form | 14.10 ± 1.20 <br> 99.2 <br> 0.6 |
| C/N rate | 2.74 |
| Total phosphor (P), % | 0.25 ± 0.06 |
| Total potassium (K), % | 0.11 ± 0.02 |
| Total sulphur (S), % | 1.10 ± 0.20 |
| Calcium (Ca), % | 0.60 ± 0.10 |
| Magnesium (Mg), % | 0.02 ± 0.003 |

The soil in the orchard was Epicalcari-Endohypogleyic Cambisol, a heavy clay loam containing 3.22% of humus, 244 mg/kg $P_2O_5$, 188 mg/kg $K_2O$, 7412 mg/kg Ca, 1848 mg/kg Mg, with pH 7.0 (in 1 M KCl extract).

In Lithuania, the growing season lasts from April until November. This period has the greatest influence on processes in plants and the soil. The perennial average precipitation during the April–November period is 464.6 mm, and the average temperature is 11.1 °C (Table 2). In 2015 and 2016, the total amount of period precipitation exceeded the perennial average, but in 2017 and 2018, it was less than normal. The average temperature of the period was higher than the perennial average in all research years.

**Table 2.** Precipitation and average air temperature during the April–November period. Records of the iMETOS® meteorological station in Babtai.

| Month | Precipitation, mm | | | | | Average Air Temperature, °C | | | | |
|---|---|---|---|---|---|---|---|---|---|---|
| | Multiannual * Average | Deviation from Multiannual Average, ± | | | | Multiannual Average | Deviation from Multiannual Average, ± | | | |
| | | 2015 | 2016 | 2017 | 2018 | | 2015 | 2016 | 2017 | 2018 |
| April | 38.4 | +33.6 | +11.8 | −8.2 | +9.8 | 6.1 | +1.6 | −0.5 | +2.9 | +1.4 |
| May | 53.8 | +6.4 | +58.6 | +49.0 | +9.4 | 12.3 | +1.4 | +3.9 | +1.3 | −0.3 |
| June | 62.6 | +28.0 | −23.6 | +22.2 | −35.8 | 15.6 | −0.1 | +3.4 | −0.8 | +0.2 |
| July | 81.2 | +19.6 | −15.2 | −30.0 | +10 | 17.6 | +2.1 | +1.4 | +2.8 | +0.8 |
| August | 80.3 | −9.9 | −2.9 | −57.5 | −75.7 | 16.6 | +0.4 | +1.6 | +6.6 | +3.1 |
| September | 52.6 | −1.8 | +31.4 | −52.6 | −9.2 | 12.2 | +1.3 | +0.1 | −0.1 | +1.8 |
| October | 49.6 | +4.4 | −5.8 | −2.6 | −38.6 | 6.8 | +0.4 | +1.9 | +0.5 | −1.2 |
| November | 46.1 | +4.9 | −1.3 | −20.1 | +26.5 | 1.5 | +3.5 | +3.6 | +1.5 | +3.5 |
| Total period | 464.6 | +85.2 | +53.0 | −99.8 | −103.6 | 11.1 | +1.3 | +1.9 | +1.8 | +1.2 |

* Multiannual average precipitation and temperature data from Lithuanian Hydrometeorological Service station in Noreikiškės, 53367 Kaunas dist., Lithuania.

Mineral nitrogen ($N_{min}$) content was determined in April, June, August and October as a sum of ammonium ($N-NH_4$) and nitrate ($N-NO_3$) ones. Soil sampling was performed in each experimental plot close to the edge of canopy projection of each five fruit trees. Five soil subsamples were mixed, and a composite soil sample of 0.5 kg was used for analysis. $N_{min}$ was established in two layers of the soil (0–30 and 31–60 cm) and presented as the average content in the 0–60 cm layer. $N-NO_3$ was ascertained via the colorimetric method using hydrazine sulphate and sulphanilamide, and $N-NH_4$ was ascertained via the colorimetric method using natrium phenolate and natrium hypochlorite.

Sampling procedure for the leaf chemical analysis was done in the first half of August. A total of 1–2 fully developed leaves were taken from the middle of current-season terminal shoots located on different sides of tree canopy at the height approximately 1.5 m. Samples of 50 leaves from each experimental plot were collected. The leaf N content was measured via the Kjeldahl method using a DK 20 Tecator Digestion System DK 20 (VelP Scientifica, Usmate, Italy) and a UDK139 Semi-Automatic Distillation Unit (VelP Scientifica, Usmate, Italy).

All laboratory analyses were performed at the Agrochemical Research Laboratory of the Lithuanian Research Center for Agriculture and Forestry, which is accredited accord-ing to the standard LST EN ISO/IEC 17025: 2018. The results of laboratory tests are pre-sented with an average of at least two replicates, with estimating the repeatability limit of the method.

Yield in each experimental plot was recorded and recalculated as tons per hectare. Yield efficiency was calculated as a ratio of yield per tree (kg) to trunk cross-section area (TCSA, $cm^2$, 20 cm above graft union) and expressed in kg $cm^2$.

Random samples of fifty apples per plot were used to determine individual fruit weight. Laboratory measurements were conducted on random samples of ten apples from each experimental plot. Fruit flesh firmness ($kg/cm^2$) was measured using a penetrometer (FT-327, TR Turoni, Forli, Italy) with an 11-mm diameter probe; soluble solids content (SSC) (% Brix) was measured using a digital refractometer (ATAGO 101, Atago Co., Ltd., Tokyo, Japan). The index of the absorption difference (IAD) between 670 and 720 nm was assessed in 2017 and 2018 using a DA-Meter (Bologna, Italy).

The data was analysed using the analysis of variance (ANOVA) procedure, and means were separated using Duncan's multiple range tests with three probability levels ($p = 0.20$, 0.10, or 0.05).

## 3. Results

### 3.1. Soil Mineral Nitrogen

Soil mineral nitrogen content (SMNC) in the middle of April 2015 was similar in all treatments (Figure 1). In June, the highest mineral nitrogen content was found in the soil to which ammonium nitrate was applied at the rate of 100 kg/ha N. In August, the highest SMNC was found in both ammonium nitrate treatments but horn shavings increased it as well. At the end of vegetation, almost all fertilised plots had higher SMNC values than the control plot. An exception was found in the plot treated with 50 kg/ha N horn shavings in spring. Here, the SMNC was the lowest during the entire vegetation period.

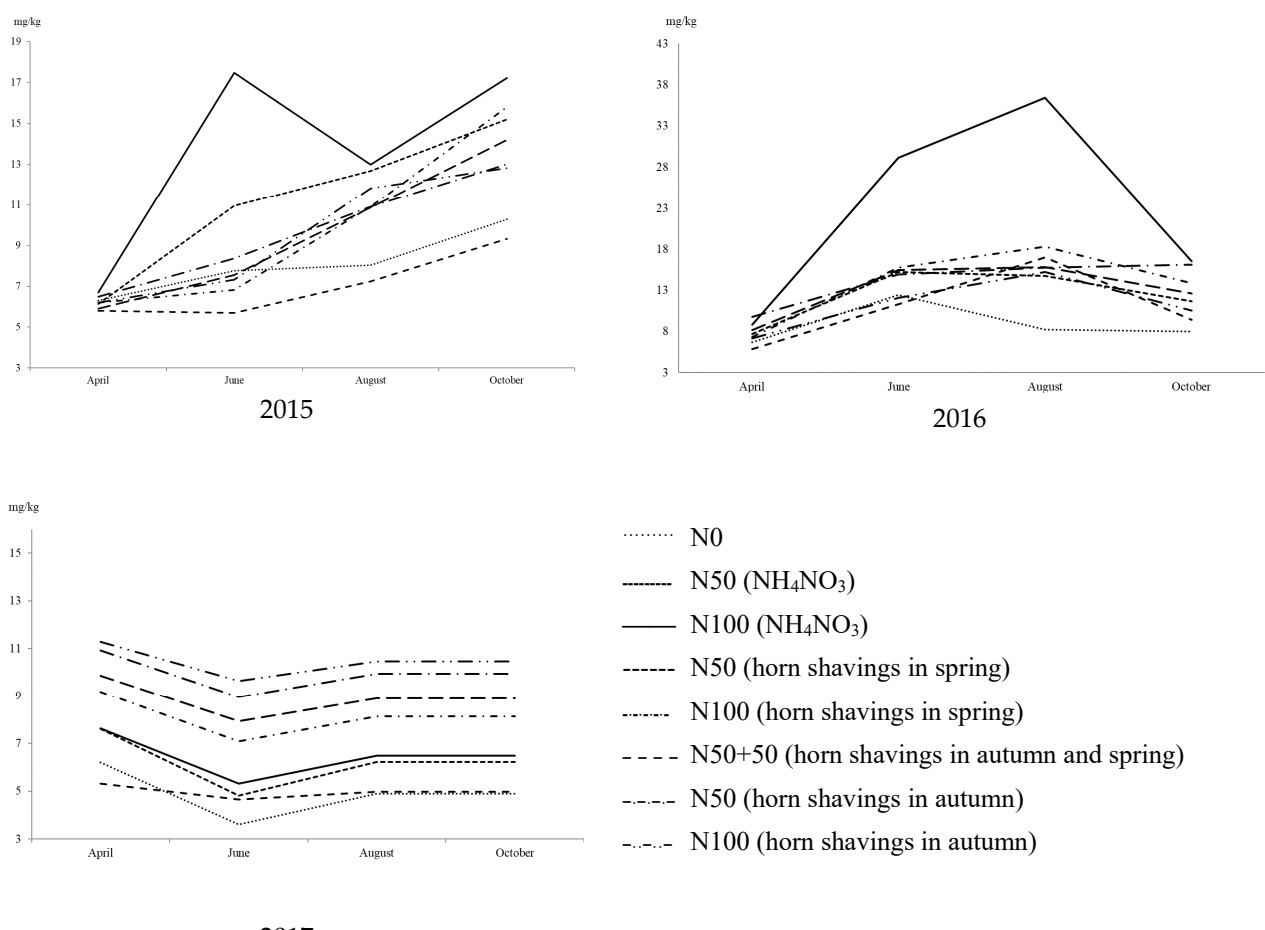

2015

2016

2017

**Figure 1.** Dynamics of mineral nitrogen content in the soil. Babtai, 2015–2017.

At the beginning of vegetation in 2016, the highest SMNC was found in the soil to which horn shavings were applied at the rate 50 kg/ha N in autumn. In treatment with the same horn shavings rate applied in spring, the SMNC was the lowest. After the application of 100 kg/ha N in the form of ammonium nitrate, the SMNC increased markedly. In most cases, horn shavings produced similar results to those of ammonium nitrate applied at the rate 50 kg/ha N. Horn shavings applied in spring at the rate of 50 kg/ha N had the smallest effect. At the end of vegetation, the highest SMNC was in the soil fertilised with 100 kg/ha N of ammonium nitrate and 50 kg/ha N of horn shavings applied in autumn. The lowest average SMNC during the vegetation period was found in the control plot, which received no fertilisers.

In the spring of the third year, the highest SMNC was recorded in almost all treatments to which horn shavings were applied, except in the case of 50 kg/ha N horn shavings applied in spring. In this treatment SMNC remained the lowest during the whole vegetation period. In the remaining treatments, where horn shavings and ammonium nitrate were applied at the rate 100 kg/ha N SMNC increased significantly. In general, in the third year SMNC was the lowest during the study.

### 3.2. Leaf Nitrogen Content

The leaf nitrogen content (LNC) of the fertilised apple trees was similar to that of the control treatment during the first year of the experiment (Table 3). The highest LNC (2.17%) was found in apple trees fertilised with 100 kg/ha N of ammonium nitrate. This content was significantly higher than that found in the leaves of apple trees fertilised with 100 kg/ha N of horn shavings in autumn (2.08%). Further nitrogen nutrition differences emerged during the second year of the experiment. The lowest LNC was detected in

control fruit trees and those fertilised with horn shavings at a rate of 50 kg/ha N in spring (1.90% and 1.95%, respectively). In other treatments, the LNC was significantly higher than in the control ($p$ = 0.05–0.2). In the third year, the highest LNC (2.17%) was found in apple trees fertilised with ammonium nitrate at the rate 100 kg/ha N. A statistically similar LNC value was found in fruit trees fertilised with horn shavings at the rate 100 kg/ha. Horn shavings used in autumn and spring had similar effects. In the final year, differences in the apple nitrogen nutrition were the most evident. The lowest LNC (1.58%) was found in the control apple trees, and horn shavings applied in spring at the rate of 50 kg/ha N had no effect. In other cases, horn shavings increased apple tree leaf nitrogen content. Their effect was similar to that of ammonium nitrate.

**Table 3.** Nitrogen content in Ligol apple tree leaves (%).

| Treatment | Year | | | |
|---|---|---|---|---|
| | **2015** | **2016** | **2017** | **2018** |
| N0 | 2.13 ab * | 1.90 a | 2.06 a | 1.58 a |
| N50 ($NH_4NO_3$) | 2.14 ab | 2.11 d | 2.12 ab | 1.89 b |
| N100 ($NH_4NO_3$) | 2.17 b | 2.07 bcd | 2.17 b | 1.86 b |
| N50 (horn shavings in spring) | 2.13 ab | 1.95 ab | 2.11 ab | 1.61 a |
| N100 (horn shavings in spring) | 2.16 b | 2.05 bcd | 2.16 b | 1.79 b |
| N50+50 (horn shavings in autumn and spring) | 2.16 b | 2.06 bcd | 2.08 a | 1.77 b |
| N50 (horn shavings in autumn) | 2.14 ab | 2.07 bcd | 2.10 a | 1.80 b |
| N100 (horn shavings in autumn) | 2.08 a | 2.08 bcd | 2.13 b | 1.82 b |
| $p$ | 0.2 | 0.2 | 0.2 | 0.05 |

* values denoted with the same letter in columns do not differ significantly at probability levels indicated in the last row of the table.

### 3.3. Apple Tree Productivity

In the first year of the experiment fertilisers had no effect on the apple yield (Table 4). In the second year, apple trees fertilised in autumn with horn shavings at the rate 50 kg/ha N and 100 kg/ha N showed the highest productivity (38.0 and 39.1 t/ha, respectively). Horn shavings applied in spring at the rate of 50 kg/ha N equivalent had no effect on apple tree productivity. In the third year, no differences in yield were found between treatments. In the final year, almost all treatments had a significant effect on the fruit tree yield ($p$ = 0.05–0.2). Only horn shavings applied in spring at the rate 50 kg/ha N were ineffective. The average yield data showed results similar to those of the final experiment year.

**Table 4.** Fertiliser effect on Ligol apple tree yield (t/ha).

| Treatment | Year | | | | Average |
|---|---|---|---|---|---|
| | **2015** | **2016** | **2017** | **2018** | |
| N0 | 33.8 a * | 26.3 a | 60.9 a | 16.9 a | 34.5 a |
| N50 ($NH_4NO_3$) | 34.8 a | 33.6 ab | 54.6 a | 46.6 c | 42.4 c |
| N100 ($NH_4NO_3$) | 33.3 a | 36.4 ab | 56.9 a | 36.7 bc | 40.8 ab |
| N50 (horn shavings in spring) | 38.1 a | 24.7 a | 62.8 a | 20.9 a | 36.6 ab |
| N100 (horn shavings in spring) | 32.9 a | 32.5 ab | 59.2 a | 35.4 abc | 40.0 abc |
| N50+50 (horn shavings in autumn and spring) | 32.7 a | 34.9 ab | 60.9 a | 31.5 abc | 40.0 abc |
| N50 (horn shavings in autumn) | 31.9 a | 38.0 b | 57.2 a | 38.4 bc | 41.4 bc |
| N100 (horn shavings in autumn) | 31.6 a | 39.1 b | 56.0 a | 31.9 abc | 39.7 bc |
| $p$ | ns ** | 0.1 | ns | 0.05 | 0.2 |

* values denoted with the same letter in columns do not differ significantly at probability levels indicated in the last row of the table. ** not significant.

Yield efficiency among treatments varied within the years. In the first year the highest yield efficiency was recorded after 50 kg/ha N equivalent horn shavings application in spring (Table 5). In the last year of the experiment fruit trees in this treatment had one of the lowest yield efficiencies similar to that in the control treatment. The last year data

highlighted the positive effects of both ammonium nitrate doses and horn shavings at 50 kg/ha N equivalent applied in autumn. The least average yield efficiency was in control treatment, the highest one—after ammonium nitrate application. Horn shavings in most cases had similar effect to the ammonium nitrate.

**Table 5.** Fertiliser effect on Ligol apple tree yield efficiency (kg/cm$^2$ TCSA).

| Treatment | Year | | | | Average |
|---|---|---|---|---|---|
| | **2015** | **2016** | **2017** | **2018** | |
| N0 | 1.44 ab * | 0.80 a | 1.37 ab | 0.34 a | 0.99 a |
| N50 (NH$_4$NO$_3$) | 1.44 ab | 1.12 ab | 1.28 ab | 0.93 b | 1.19 b |
| N100 (NH$_4$NO$_3$) | 1.41 ab | 1.23 b | 1.32 ab | 0.74 b | 1.18 b |
| N50 (horn shavings in spring) | 1.64 b | 0.97 ab | 1.43 b | 0.40 a | 1.11 ab |
| N100 (horn shavings in spring) | 1.38 ab | 1.11 ab | 1.37 ab | 0.71 ab | 1.14 ab |
| N50+50 (horn shavings in autumn and spring) | 1.28 ab | 1.02 ab | 1.27 ab | 0.57 ab | 1.04 ab |
| N50 (horn shavings in autumn) | 1.30 ab | 1.17 b | 1.27 ab | 0.73 b | 1.12 ab |
| N100 (horn shavings in autumn) | 1.19 a | 1.16 b | 1.18 a | 0.59 ab | 1.03 ab |
| *p* | 0.1 | 0.2 | 0.2 | 0.05 | 0.05 |

* values denoted with the same letter in columns do not differ significantly at probability levels indicated in the last row of the table.

### 3.4. Fruit Quality

The average fruit weight at the beginning of the experiment was similar in all treatments. The differences appeared in the second year of the experiment, but the studied fertilizers and their application time did not have a regular effect on this indicator. Average apple weight was 232–250 g (Table 6). The largest fruits were produced by apple trees fertilised with horn shavings in autumn at the rate 50 kg/ha N, and the smallest ones were produced by the most productive fruit trees, which were fertilised with ammonium nitrate at the rate of 50 kg/ha N.

**Table 6.** Fertiliser effect on Ligol apple average fruit weight (g).

| Treatment | Year | | | | Average |
|---|---|---|---|---|---|
| | **2015** | **2016** | **2017** | **2018** | |
| N0 | 257 | 265 ab * | 235 ab | 234 bc | 248 ab |
| N50 (NH$_4$NO$_3$) | 234 | 244 a | 239 ab | 210 ab | 232 a |
| N100 (NH$_4$NO$_3$) | 260 | 275 b | 222 a | 208 ab | 241 ab |
| N50 (horn shavings in spring) | 242 | 261 ab | 221 a | 210 ab | 234 a |
| N100 (horn shavings in spring) | 260 | 240 a | 247 b | 198 a | 236 ab |
| N50+50 (horn shavings in autumn and spring) | 260 | 276 b | 228 ab | 187 a | 238 ab |
| N50 (horn shavings in autumn) | 257 | 265 ab | 233 ab | 244 c | 250 b |
| N100 (horn shavings in autumn) | 270 | 258 ab | 236 ab | 219 bc | 246 ab |
| *p* | ns ** | 0.05 | 0.05 | 0.05 | 0.05 |

* values denoted with the same letter in columns do not differ significantly at probability levels indicated in the last row of the table. ** not significant.

Apple flesh firmness at the beginning of the experiment was similar in all treatments. From the second year of the study, statistical differences occurred, but there was no constant effect of different fertilizers: fruit flesh firmness from the trees fertilised with horn shavings 100 kg/ha N equivalent varies from the lowest to the highest in different years (Table 7). In the last year the firmest apple flesh was in the control treatment. From the average data, apples from plots fertilised with horn shavings at 50 kg/ha N equivalent in spring had the highest flesh firmness (7.2 kg/cm$^2$). Apple trees fertilised with 100 kg/ha N equivalent of ammonium nitrate and horn shavings, both in spring and in equal parts in spring and autumn, produced fruits with somewhat softer flesh (6.9 kg/cm$^2$).

**Table 7.** Fertiliser effect on Ligol apple flesh firmness (kg/cm$^2$).

| Treatment | Year | | | | Average |
|---|---|---|---|---|---|
| | **2015** | **2016** | **2017** | **2018** | |
| N0 | 7.6 | 7.8 ab * | 6.1 ab | 6.9 b | 7.1 ab |
| N50 (NH$_4$NO$_3$) | 7.8 | 8.1 ab | 6.1 ab | 6.3 ab | 7.1 ab |
| N100 (NH$_4$NO$_3$) | 7.6 | 7.7 ab | 5.9 a | 6.5 ab | 6.9 a |
| N50 (horn shavings in spring) | 7.6 | 8.3 b | 6.2 ab | 6.7 ab | 7.2 b |
| N100 (horn shavings in spring) | 7.9 | 7.6 a | 6.3 b | 6.0 a | 6.9 a |
| N50+50 (horn shavings in autumn and spring) | 7.6 | 7.6 ab | 6.1 ab | 6.2 ab | 6.9 a |
| N50 (horn shavings in autumn) | 7.7 | 7.7 ab | 6.3 b | 6.5 ab | 7.1 ab |
| N100 (horn shavings in autumn) | 7.8 | 7.6 ab | 6.3 b | 6.4 ab | 7.0 a |
| *p* | ns ** | 0.05 | 0.05 | 0.1 | 0.05 |

\* values denoted with the same letter in columns do not differ significantly at probability levels indicated in the last row of the table. \*\* not significant.

In the first year of the experiment the lowest soluble solids content (SSC) had fruits from the control treatment (Table 8). In most of treatments with horn shavings application SSC was significantly higher. The second and the third years did not reveal differences among treatments. In the last year, the highest SSC was in fruits after 50 kg/ha N equivalent horn shavings application in spring, the lowest one—In the control treatment. On the average data, the lowest SSC was found in fruits from the control treatment (12.6%). Fruits from apple trees fertilised with ammonium nitrate and, in many cases, with horn shavings had higher SSC (13.1–13.4%).

**Table 8.** Fertiliser effect on Ligol apple soluble solids content (SSC, %).

| Treatment | Year | | | | Average |
|---|---|---|---|---|---|
| | **2015** | **2016** | **2017** | **2018** | |
| N0 | 12.0 a * | 13.6 | 12.8 | 12.1 a | 12.6 a |
| N50 (NH$_4$NO$_3$) | 12.5 ab | 14.2 | 13.5 | 12.9 ab | 13.3 bc |
| N100 (NH$_4$NO$_3$) | 12.5 ab | 13.8 | 12.7 | 13.4 bc | 13.1 b |
| N50 (horn shavings in spring) | 12.2 a | 14.3 | 12.8 | 14.3 c | 13.4 c |
| N100 (horn shavings in spring) | 12.7 b | 13.7 | 12.7 | 13.9 bc | 13.2 bc |
| N50+50 (horn shavings in autumn and spring) | 12.8 b | 13.3 | 12.9 | 12.5 ab | 12.9 ab |
| N50 (horn shavings in autumn) | 12.9 b | 13.6 | 12.6 | 12.5 ab | 12.9 ab |
| N100 (horn shavings in autumn) | 12.9 b | 13.4 | 13.0 | 13.2 b | 13.1 b |
| *p* | 0.05 | ns ** | ns | 0.05 | 0.1 |

\* values denoted with the same letter in columns do not differ significantly at probability levels indicated in the last row of the table. \*\* not significant.

IAD can be used to estimate the maturity and storability of apples [17]. In both years of the experiment when IAD was measured the lowest values were recorded in the control treatment (Table 9). On the average data, low IAD values were measured in apples from plots fertilised with horn shavings at the rate of 50 kg/ha N equivalent in spring (0.57 and 0.76, respectively). The rest of the fertilised treatments, including those with ammonium nitrate and horn shavings, increased IAD values.

In the first year of the experiment the fruits with the best colour were from trees fertilised with 100 kg/ha N equivalent of horn shavings in equal parts in autumn and spring (Table 10). There was no difference in fruit colour in the second year. In the third and fourth years, fruits in the control treatment were the most intensely covered with red blush (72 and 59%, respectively). In the final year, the fruits of apple trees fertilised with horn shavings in spring at 50 kg/ha N equivalent had better colour as well.

**Table 9.** Fertiliser effect on index of the absorption difference (IAD) of Ligol apple.

| Treatment | Year | | Average |
|---|---|---|---|
| | **2017** | **2018** | |
| N0 | 0.83 a * | 0.31 a | 0.57 a |
| N50 (NH$_4$NO$_3$) | 1.11 b | 0.80 bc | 0.96 b |
| N100 (NH$_4$NO$_3$) | 1.14 b | 0.94 c | 1.04 b |
| N50 (horn shavings in spring) | 0.98 ab | 0.53 ab | 0.76 ab |
| N100 (horn shavings in spring) | 1.12 b | 0.80 bc | 0.96 b |
| N50+50 (horn shavings in autumn and spring) | 1.03 ab | 0.94 c | 0.98 b |
| N50 (horn shavings in autumn) | 1.12 b | 0.73 bc | 0.92 b |
| N100 (horn shavings in autumn) | 0.98 ab | 0.77 bc | 0.88 b |
| *p* | 0.05 | 0.05 | 0.05 |

* values denoted with the same letter in columns do not differ significantly at probability levels indicated in the last row of the table.

**Table 10.** Fertiliser effect on Ligol apple colour (%).

| Treatment | Year | | | |
|---|---|---|---|---|
| | **2015** | **2016** | **2017** | **2018** |
| N0 | 58 a * | 68 a | 72 b | 59 b |
| N50 (NH$_4$NO$_3$) | 59 ab | 66 a | 65 a | 51 a |
| N100 (NH$_4$NO$_3$) | 63 bc | 69 a | 65 a | 47 a |
| N50 (horn shavings in spring) | 61 ab | 69 a | 66 a | 56 ab |
| N100 (horn shavings in spring) | 61 ab | 65 a | 65 a | 45 a |
| N50+50 (horn shavings in autumn and spring) | 66 c | 68 a | 67 ab | 48 a |
| N50 (horn shavings in autumn) | 58 a | 64 a | 67 ab | 41 a |
| N100 (horn shavings in autumn) | 62 b | 63 a | 67 ab | 44 a |
| *p* | 0.1 | ns ** | 0.1 | 0.1 |

* values denoted with the same letter in columns do not differ significantly at probability level indicated in the last row of the table. ** not significant.

## 4. Discussion

### 4.1. Soil Mineral Nitrogen

Our research shows that horn shavings increased SMNC in many cases. Horn shavings applied at rates of 50 and 100 kg/ha N equivalent produced results similar to those of 50 kg/ha N of ammonium nitrate. In most cases, the highest rate of ammonium nitrate (100 kg/ha N) increased SMNC rapidly, and the increase was more evident. However, horn shavings are a slow-release nitrogen fertiliser [10]. SMNC and the effect of fertilisers on it depended on the year conditions. The lowest SMNC was found in 2017, when the amount of precipitation during the vegetation period was about 20% lower than the multiannual average. Similar observations were reported in other studies with horn shavings [18]. Despite the dry vegetation season in 2017, May and June were wet. This could lead to nitrogen leaching in plots where ammonium nitrate was applied, as well as to relatively low SMNC. Application time (spring or autumn) was important when 50 kg/ha N equivalent of horn shavings was applied. Horn shavings applied at this rate in spring resulted in lower SMNC. The mineralisation of organic nitrogen compounds mainly includes the biological process affected by temperature and soil humidity [19,20]. The process of cattle horn shavings mineralization starts more slowly at a lower temperature, but later the process accelerates even under the same conditions [21]. Despite the more favourable temperatures, the soil surface is usually drier in spring, and the preconditions for mineralisation become less favourable.

Simulating technological applicability in commercial orchards, fertilisers were broadcasted throughout the entire area of experimental plots on the soil surface in our experiment. Considering the assumptions of environmental conditions for the mineralization process,

incorporation of horn shavings into the soil could improve their mineralisation and increase efficiency.

### 4.2. Leaf Nitrogen Content

The effect of fertilisers effect on LNC began to be noticeable in the second year of the experiment. Sometimes, even if the nutrient availability is below the minimum threshold, trees do not respond to fertilisation because of adequate nutrient reserves built up in perennial organs during previous years [22,23]. The most obvious differences in LNC were recorded only in the fourth year of the experiment. The delayed influence of nitrogen fertilisers on LNC is also mentioned in other studies [24]. Both ammonium nitrate and horn shavings increased LNC in our experiment. As in the case of SMNC, 50 kg/ha N equivalent of horn shavings applied in spring did not influence effectively LNC. Overall, in 2018, LNC was evaluated as low or deficient [25]. This could be due to dry, warm weather conditions.

### 4.3. Apple Tree Productivity

The effect of fertilisers on apple yield was not immediate. The highest yield differences emerged in the final year of the experiment when control trees and those what received 50 kg/ha N equivalent of horn shavings in spring produced the lowest yield. The yield in the rest N treatments was greater. Almost identical trend of yield efficiency was observed. Similar information about positive but delayed N effect on yield and yield efficiency was observed in other studies [26].

### 4.4. Fruit Quality

Better nitrogen nutrition had no significant effect on average fruit weight, and non-fertilised fruit trees produced rather large fruits. This may be associated with lower yields: trees that produce fewer fruits also produce larger fruits [27].

The nitrogen fertilisers only slightly affected fruit flesh firmness, and higher fertiliser rates had a greater impact. Racsko et al. [28] also reported the negative effects of 100 kg/ha N fertiliser rate on apple flesh firmness Nitrogen fertilisers also slightly increased SSC in our experiment. Information about nitrogen effect on this fruit quality indicator is different in other studies. Nava et al. [29] reported a negative effect of nitrogen fertilisation on SCC, whereas, according to Jivan and Sala [30] higher LNC may slightly increase or decrease SSC.

IAD was measured during the last two years of the experiment. It indicates chlorophyll content in the external fruit tissues [31]. IAD is a reliable parameter for assessing fruit maturity at harvest [32]. Decreasing index values correspond to increasingly advanced stages of the ripening process [33]. In our experiment, almost all nitrogen applications increased IAD, which indicates delayed fruit ripening. Only horn shavings used in spring at 50 kg/ha N equivalent had no effect on IAD. Like IAD, fruit colour changes depending on nitrogen nutrition were observed. In the first year of the experiment, fruits from unfertilised apple trees were less covered with the red blush compared to those from most treatments supplied with nitrogen fertilisers. The last two years revealed that apple trees supplied with the nitrogen produced fewer coloured fruits. This confirmed the fact that increasing N supply delays skin red colour development [34].

## 5. Conclusions

In summary, the effects of horn shavings on apple trees in an orchard were equivalent to those of ammonium nitrate in many cases. This organic fertiliser of animal origin increased SMNC, improved apple trees nitrogen nutrition and increased their yield. However, horn shavings like ammonium nitrate slightly delayed fruit ripening and reduced their coloration. Cattle horn shavings could be used to manage nitrogen nutrition in apple trees especially in organic orchards.

**Author Contributions:** Conceptualization, methodology, investigation and writing: J.L.; technological supervision of field experiment and investigation: N.U.; data collection: L.B.; soil testing and data evaluation: G.S.; leaf analysis and data evaluation: R.M.; administration and funding acquisition: D.K. All authors have read and agreed to the published version of the manuscript.

**Funding:** This research was funded by long-term science programme "Horticulture: agrobiological foundations and technologies", approved by order No V-273 of Minister of Education and Science of the Republic of Lithuania.

**Institutional Review Board Statement:** Not applicable.

**Informed Consent Statement:** Not applicable.

**Conflicts of Interest:** The authors declare no conflict of interest.

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
