# Peer review of "Cattle Horn Shavings: A Possible Nitrogen Source for Apple Trees"

_agronomy, doi:10.3390/agronomy11030540_

Round 1
Reviewer 1 Report
Dear Authors,
In reviewed manuscript ‘Cattle horn shavings: A possible nitrogen source for apple trees’ Authors made attempt to investigate the effects of cattle horn shavings on apple tree nitrogen nutrition.
It has to be deeply revised and resubmitted. The greatest disadvantage of this manuscript is English, the quality of which is low. The article is difficult to understand in many places. Paper is also very simple and chaotic.
Introduction
The introduction is very short and simple. Little can be learned about nitrogen fertilization and the use of waste from animal production for this purpose. The authors could write something more about the mineralization of organic nitrogen in the soil. Is there anything more known about nitrogen release from cattle horn shavings? Are other products after animal production used in nitrogen fertilization? What are the results? Do the cattle horn shavings have any disadvantages and dangers when are used as nitrogen source?
Materials and methods
The materials and methods are also not clear. Whole section is difficult to understand and not properly structured.
How many experimental plots per combination were used?
The whole paragraph (lines 72-77) has to be revised. There should be no doubts how fertilizers were applied. Maybe table would help in presentation of experimental combinations? How technically cattle horn shavings were applied?
Please use subscript in chemical formulas and everywhere else where required.
Line 82: by perennial you mean annual?
Lines 89-96: how many subsamples of soil was in one samples? What is ‘Nmin’?
Could you show number of replications in each laboratory test? This should also be indicated in graphs and tables?
Results are mainly simple presentation of individual values for graph and tables.
Lines 113-142: Even If there is only one graph, it should have number. Is it possible, please do one graph out of three? Are the changes significant? There is no need to write so many details, all is presented on graphs. Please describe only most important observation.
Lines 141-143 – this is discussion.
Tables 2-3: why not use only probability levels 0,05?
Discussion is very simple and chaotic. Authors have to revise it, think about own findings and explanations. Maybe it would be good to divide the discussion into subsections.
with best regards
Author Response
In reviewed manuscript ‘Cattle horn shavings: A possible nitrogen source for apple trees’ Authors made attempt to investigate the effects of cattle horn shavings on apple tree nitrogen nutrition.
It has to be deeply revised and resubmitted. The greatest disadvantage of this manuscript is English, the quality of which is low. The article is difficult to understand in many places. Paper is also very simple and chaotic.
Some words about the quality of the English language. The text of the first submission was evaluated by English language specialists from https://www.proof-reading-service.com/. We are not experts in English, so we trusted the quality of their work. We apologize for the incorrect English language and promise to seek qualified assistance in this matter if you are satisfied with the content of the article.
Introduction
The introduction is very short and simple. Little can be learned about nitrogen fertilization and the use of waste from animal production for this purpose. The authors could write something more about the mineralization of organic nitrogen in the soil. Is there anything more known about nitrogen release from cattle horn shavings? Are other products after animal production used in nitrogen fertilization? What are the results? Do the cattle horn shavings have any disadvantages and dangers when are used as nitrogen source?
“Introduction” was supplemented with the following information on the use of animal waste as a source of mineral nutrients:
According to the requirements of EU regulations [12], animal by-products can be used in the organic farming system. Products of animal origin are used to fertilize plants, including keratin-containing waste: horn chips, shredded feathers, bristles, horn core powder, etc. [13, 14]. Keratin contains about 15–18% nitrogen, 1.5–2.0% phosphorus, sulfur and other elements that are in organic form. Keratin-containing wastes decompose more slowly than other organic animal wastes because of cysteine, sulfur-containing amino acids, that form strong intermolecular bonds and give to the protein a crystalline structure and strength. Cattle horns used to be a great raw material for haberdashery, but recently they have been replaced by more colourful and easier to recycle plastics.
More than half of the animal by-products are not suitable for normal consumption, because of their unusual physical and chemical characteristics [15]. Slaughterhouses generate between 0.5 and 2 kg of horns and other keratin-containing waste from each slaughtered bovine animal. As a result, thousands of tons of waste are accumulated each year that could be used in organic horticulture.”
Materials and methods
The materials and methods are also not clear. Whole section is difficult to understand and not properly structured.
“Materials and methods” have been supplemented and modified. Details about the changes are set out after specific comments.
How many experimental plots per combination were used?
Additional information provided in the manuscript: “The field experiment was arranged in randomised block design, and had four replications.”
The whole paragraph (lines 72-77) has to be revised. There should be no doubts how fertilizers were applied. Maybe table would help in presentation of experimental combinations? How technically cattle horn shavings were applied?
Additional information provided in the manuscript:
“Scheme of the experiment:
- No fertilisers
- 50 kg/ha N equivalent applied in spring as NH4NO3
- 100 kg/ha N equivalent applied in spring as NH4NO3
- 50 kg/ha N equivalent applied in spring as horn shavings
- 100 kg/ha N equivalent applied in spring as horn shavings
- 50+50 N equivalent applied at equal parts in autumn and spring as horn shavings
- 50 kg/ha N equivalent applied in autumn as horn shavings
- 100 kg/ha N equivalent applied in autumn as horn shavings
In spring fertilizers were applied at 07-09 BBCH, in autumn – at 95 BBCH stage [16]. The first autumn application of horn shavings was performed in 2014. The effects of the horn shavings were compared to the effects of ammonium nitrate (34.4% N) and unfertilised treatment. Both fertilisers were broadcasted throughout the entire area of the relevant experimental plots on the soil surface.”
Please use subscript in chemical formulas and everywhere else where required.
The specified inaccuracies have been corrected.
Line 82: by perennial you mean annual?
Instead of “Perenial” we changed into “Multiannual”.
Lines 89-96: how many subsamples of soil was in one samples? What is ‘Nmin’?
Lines were modified, required information was presented:
“Mineral nitrogen (Nmin) content was determined in April, June, August and October as a sum of ammonium (N-NH4) and nitrate (N-NO3) ones. Soil sampling was performed in each experimental plot close to the edge of canopy projection of each fruit tree. 5 soil subsamples were mixed, and a composite soil sample of 0.5 kg was used for analysis.”
Could you show number of replications in each laboratory test? This should also be indicated in graphs and tables?
The following information was provided:
“All laboratory analyzes were performed at the Agrochemical Research Laboratory of the Lithuanian Research Center for Agriculture and Forestry, which is accredited according to the standard LST EN ISO/IEC 17025: 2018. The results of laboratory tests are presented with an average of at least two replicates, with estimating the repeatability limit of the method.”
We think that biological replicates in situ are more significant because the accuracy of laboratory tests usually is enough high.
Results are mainly simple presentation of individual values for graph and tables.
We don’t really understand what this remark means.
Lines 113-142: Even If there is only one graph, it should have number. Is it possible, please do one graph out of three? Are the changes significant? There is no need to write so many details, all is presented on graphs. Please describe only most important observation.
The number of figure was indicated. We consider it is appropriate to leave annual graphs of soil Nmin content dynamics as the differences between the years are obvious and possibly related to the meteorological conditions. It is difficult to indicate significance because graphs would contain a lot of information and become difficult to understand.
Corrections in result presentation were done eliminating redundant information.
Lines 141-143 – this is discussion.
Information deleted, issues are discussed in the section “Discussion”.
Tables 2-3: why not use only probability levels 0,05?
Short explanation why we use not only 0.05 but also a lower probability level when evaluating experimental data. Changes in the mineral nutrition of fruit trees are inert and trends take some time to emerge. So, we think it is worth pointing out the significance of some differences even at a lower level of probability (p=0.1 or 0.2).
Discussion is very simple and chaotic. Authors have to revise it, think about own findings and explanations. Maybe it would be good to divide the discussion into subsections.
“Discussion” was modified and divided into subsections.
Reviewer 2 Report
The article reports the effects of fertilization with cattle horn shavings on productivity and on some qualitative characteristics of apple fruit. In particular, the effects of two doses (50 and 100 kg ha N) and of two application periods (end of autumn or beginning of vegetation in spring) of a horn shavings fertilizer with the effects of ammonium nitrate fertilizer and unfertilized treatment were compared. While not highly novel, it contributes to increasing knowledge about manage nitrogen nutrition management in apple trees, especially in organic orchards, where the use of synthetic fertilisers is prohibited. However there are deficiencies in trial description and presentation of results that make it difficult to judge the importance and real value of the study and therefore should be substantially revised. The paper would benefit from moderate editing for grammar and spelling.
Major comments:
MATERIALS AND METHODS:
The materials and methods lack information on the implementation of the experiment which precludes the reproducibility of the study.
Lines 68 and 69. The experimental design is not clear. The manuscript talks about blocks of five plants distributed with completely randomized design but it is not clear how many blocks the treatment was repeated. Please describe this point better.
Line 71. It is reported that the fruits have not been thinned. Explain the reasons for this choice.
Line 72. With regard to the cattle horn shavings used in the experiment, the dimensions and chemical composition of cattle horn shavings must be indicated. Insert a table showing this information.
Lines 72 and 73. With regard to the time of application of fertilizers, in addition to the period, the corresponding phenological stage of the plants must also be indicated.
Lines 76 and 77. Explain the reasons why cattle horn shavings were not incorporated into the soil after their application. This choice certainly influenced the effectiveness of this fertilizer, partly masking the results obtained in this study.
Lines 82-86. With regard to the thermopluviometric data, it should be indicated where the historical data and those relating to the years of experimentation were acquired. Considering the objectives of the study, to better understand the dynamics of mineral nitrogen content in the soil, in addition to the air temperature data, those relating to the soil temperature should also be reported. In addition to soil moisture the mineralisation of organic nitrogen compounds is influenced by soil temperature.
Lines 92-96. The manuscript reports that the Nmin content was determined in two soil layers at different depths (0-30 and 31-60 cm). Subsequently, in the results section, it is reported as the average content. Instead it would be useful to report the Nmin content in the two different layers in a distinct way. Report in a clearer and more complete way the methodology used to determine the Nmin, N-NO3 and N-NH4 contents.
Lines 97 and 98. It should be clearly indicated how the leaves for foliar analyzes were sampled, specifying the type of shoots from which the leaves were taken and the age of the leaves.
Lines 101 and 102. The harvest time must be indicated.
RESULTS:
The results should be improved and possibly integrated with other data that allow to better understand the effects of the different treatments. It would also be advisable to reconsider the way the results are presented.
Lines 165 – 173. In addition to the yield per hectare, for each year the average values of the yield, fruit number and yield efficiency (expressed as kilograms of fruit per square centimeter of trunk cross-sectional area or kilograms of fruit per square meter of canopy volume) should be reported.
Line 193. As fact for the other production parameters shown in tables 3 and 5, the data on the average fruit weight, fruit flesh firmness, soluble solids content and IAD should be reported for each year and not as an average value for the three-year period. Insert a new table showing this information.
DISCUSSION:
Please review this section on the basis of changes that have been suggested in the results section.
Author Response
Major comments:
MATERIALS AND METHODS:
The materials and methods lack information on the implementation of the experiment which precludes the reproducibility of the study.
Lines 68 and 69. The experimental design is not clear. The manuscript talks about blocks of five plants distributed with completely randomized design but it is not clear how many blocks the treatment was repeated. Please describe this point better.
Additional information provided in the manuscript: “The field experiment was arranged in randomised block design, and had four replications.”
Line 71. It is reported that the fruits have not been thinned. Explain the reasons for this choice.
Fruitlet thinning for large fruit cultivars is not common practise in Lithuanian commercial orchards. Cultivar Ligol usually produces large fruits at the young age. Our experiment was performed in 2-6-year-old orchard, and average fruit size exceeded 200 g. Under these circumstances, fruitlet thinning would not be appropriate. This could result too big fruits and decrease their storability. Fruitlet thinning for Ligol cultivar becomes actual in older orchard when fruits become smaller.
Line 72. With regard to the cattle horn shavings used in the experiment, the dimensions and chemical composition of cattle horn shavings must be indicated. Insert a table showing this information.
Required information about chemical composition and dimensions of cattle horn shavings was supplemented in “Materials and methods”:
For the research were used cattle horn shavings from a haberdashery company. The company purchases horn waste from JSC Utenos mÄ—sa (meat). The waste generated during production is shredded into horn shavings with a special mill. A 2.5-3.0 mm chip fraction was used for the research, which is similar in size to the mineral fertilizer granules, containing 14.1% N (Table 1).
Table 1. Chemical composition of Cattle horn shavings
Composition |
Content, % |
Dry matter |
91,20±1,60 |
Organic matter |
98,30±0,90 |
Organic carbon |
38,65±1,30 |
Total nitrogen (N): N in organic form, N in inorganic form |
14,10±1,20 99,2 0,6 |
C/N rate |
2,74 |
Total phosphor (P), % |
0,25±0, 06 |
Total potassium (K), % |
0,11±0,02 |
Total sulfur (S), % |
1,10±0,20 |
Calcium (Ca), % |
0,60±0,10 |
Magnesium (Mg), % |
0,02±0,003 |
Lines 72 and 73. With regard to the time of application of fertilizers, in addition to the period, the corresponding phenological stage of the plants must also be indicated.
Required information on the phenological phases of apple trees during fertilization was added:
“In spring fertilizers were applied at 07-09, in autumn – at 95 BBCH scale stage [16].”
Lines 76 and 77. Explain the reasons why cattle horn shavings were not incorporated into the soil after their application. This choice certainly influenced the effectiveness of this fertilizer, partly masking the results obtained in this study.
Fertilisers were broadcasted throughout the entire area of experimental plots on the soil surface. This method of application is technologically acceptable in most of commercial orchards. We agree with your opinion that incorporation of horn shavings into the soil should have a more pronounced effect. Further research related to the aspects of orchard soil maintenance could provide more detailed answers to these predictions. These considerations are reflected in the Discussion.
Lines 82-86. With regard to the thermopluviometric data, it should be indicated where the historical data and those relating to the years of experimentation were acquired. Considering the objectives of the study, to better understand the dynamics of mineral nitrogen content in the soil, in addition to the air temperature data, those relating to the soil temperature should also be reported. In addition to soil moisture the mineralisation of organic nitrogen compounds is influenced by soil temperature.
Historical data about temperature and precipitation were taken from meteorological station in NoreikiškÄ—s, 53367 Kaunas dist., Lithuania, Lithuanian Hydrometeorological Service. For current data registration was used iMETOS® meteorological station in experimental orchard, Babtai. Direct distance between meteorological station in NoreikiškÄ—s and Babtai is about 23 km.
We agree with your comment – soil moisture and temperature are very important factors influencing mineralisation of organic compounds. This is confirmed by the differences in the content of Nmin in the soil in different years of the study. We relied on weather temperature and precipitation because these data are comparable with multiannual records. In addition, the correlation between soil and air temperature was quite close in our case (r=0.97-0.99). We think the provided information is sufficient to illustrate the significance of temperature and humidity on the mineralization of organic matter.
Lines 92-96. The manuscript reports that the Nmin content was determined in two soil layers at different depths (0-30 and 31-60 cm). Subsequently, in the results section, it is reported as the average content. Instead it would be useful to report the Nmin content in the two different layers in a distinct way. Report in a clearer and more complete way the methodology used to determine the Nmin, N-NO3 and N-NH4 contents.
According to many observations, a large number of dwarf apple roots are up to 60 cm deep. 0-60 cm soil layer is an important source of mineral nutrients. Two soil layers were analysed separately due to the specifics of sampling: soil sampling probes for 0-30 cm and 31-60 cm were used.
Information in Materials and methods was modified for greater clarity:
“Mineral nitrogen (Nmin) content was determined in April, June, August and October as a sum of ammonium (N-NH4) and nitrate (N-NO3) ones. Soil sampling was performed in each experimental plot close to the edge of canopy projection of each fruit tree. 5 soil subsamples were mixed, and a composite soil sample of 0.5 kg was used for analysis.”
Lines 97 and 98. It should be clearly indicated how the leaves for foliar analyzes were sampled, specifying the type of shoots from which the leaves were taken and the age of the leaves.
Information about leaf sampling in Materials and methods was modified for greater clarity:
“Sampling procedure for the leaf chemical analysis was done in the first half of August. 1-2 fully developed leaves were taken from the middle of current-season terminal shoots located on different sides of tree canopy at the height approximately 1.5 m. Samples of 50 leaves from each experimental plot were collected.”
Lines 101 and 102. The harvest time must be indicated.
Needed information was added to Materials and methods: “Fruits were harvested in the second decade of October.”
RESULTS:
The results should be improved and possibly integrated with other data that allow to better understand the effects of the different treatments. It would also be advisable to reconsider the way the results are presented.
For better understanding section Results was divided into subsections: Soil mineral nitrogen, Leaf nitrogen content, Apple tree productivity and Fruit quality. Corrections in result presentation were done eliminating redundant information.
Lines 165 – 173. In addition to the yield per hectare, for each year the average values of the yield, fruit number and yield efficiency (expressed as kilograms of fruit per square centimeter of trunk cross-sectional area or kilograms of fruit per square meter of canopy volume) should be reported.
Yield efficiency as a ratio of yield per tree (kg) to trunk cross-section area (cm2) was calculated and presented.
Line 193. As fact for the other production parameters shown in tables 3 and 5, the data on the average fruit weight, fruit flesh firmness, soluble solids content and IAD should be reported for each year and not as an average value for the three-year period. Insert a new table showing this information.
The section Results has been supplemented with annual data of average fruit weight, fruit flesh firmness, soluble solids content and IAD.
DISCUSSION:
Please review this section on the basis of changes that have been suggested in the results section.
For easier reading and understanding Discussion was divided into sections (Soil mineral nitrogen, Leaf nitrogen content, Apple tree productivity and Fruit quality). Some additional information is provided.
Round 2
Reviewer 1 Report
Dear Authors,
manuscript is significantly improved.
What you still need to improve is formatting (margins, double spaces, etc.).
I do not understand this sentence in abstract: 'In one treatment, 100 kg/ha N was applied in equal parts in autumn and spring'. Please revise.
best regards
Author Response
Thanks for the comments. Below are our explanations.
What you still need to improve is formatting (margins, double spaces, etc.).
Detected formatting inaccuracies have been corrected. Some additional corrections we think will be made by the editor(s) in the final version of the publication.
I do not understand this sentence in abstract: 'In one treatment, 100 kg/ha N was applied in equal parts in autumn and spring'. Please revise.
We have made the following corrections to make the text clearer:
“Horn shavings (14.1% N) were applied at the end of autumn or at the beginning of vegetation in the spring and in one treatment 100 kg/ha N rate was divided into two equal parts and applied both in autumn and spring. In one treatment, 100 kg/ha N was applied in equal parts in autumn and spring.”
Reviewer 2 Report
The article reports the effects of fertilization with cattle horn shavings on productivity and on some qualitative characteristics of apple fruit. The topic is interesting, but not very innovative. The manuscript is overall well written and quite readable. However, the manuscript would benefit from a moderate modification to improve the English form.Author Response
Thanks for the comments. Below are our explanations.
The article reports the effects of fertilization with cattle horn shavings on productivity and on some qualitative characteristics of apple fruit. The topic is interesting, but not very innovative. The manuscript is overall well written and quite readable. However, the manuscript would benefit from a moderate modification to improve the English form.
Since we are not experts in the English language, we hope that minor corrections could made by the editor(s). We would be very grateful for that.